

# Differences of soil enzyme activities and its influencing factors under different flooding conditions in Ili Valley, Xinjiang

Yulu Zhang[1,2], Dong Cui[1,2], Haijun Yang[1,2,3] and Nijat Kasim[1]

[1] College of Biology and Geography Sciences, Yili Normal University, Yining, Yili Kazakh Autonomous Prefecture, China
[2] Institute of Resources and Ecology, Yili Normal University, Yining, Yili Kazakh Autonomous Prefecture, China
[3] Ministry of Education Key Laboratory of Vegetation Ecology, Institute of Grassland Science, Northeast Normal University, Changchun, China

## ABSTRACT

**Background**. A wetland is a special ecosystem formed by the interaction of land and water. The moisture content variation will greatly affect the function and structure of the wetland internal system.

**Method**. In this paper, three kinds of wetlands with different flooding levels (*Phragmites australis wetland* (long-term flooding), *Calamagrostis epigeios wetland* (seasonal flooding) and *Ditch millet wetland* (rarely flooded)) in Ili Valley of Xinjiang China were selected as research areas. The changes of microbial biomass carbon, soil physical and chemical properties in wetlands were compared, and redundancy analysis was used to analyze the correlation between soil physical and chemical properties, microbial biomass carbon and enzyme activities (soil sucrase, catalase, amylase and urease). The differences of soil enzyme activities and its influencing factors under different flooding conditions in Ili Valley were studied and discussed.

**Result**. The results of this study were the following: (1) The activities of sucrase and amylase in rarely flooded wetlands and seasonally flooded wetlands were significantly higher than those in long-term flooded wetlands; the difference of catalase activity in seasonal flooded wetland was significant and the highest. (2) Redundancy analysis showed that soil organic carbon, dissolved organic carbon, total phosphorus and soil microbial biomass carbon had significant effects on soil enzyme activity ($p < 0.05$). (3) The correlation between soil organic carbon and the sucrase activity, total phosphorus and the catalase activity was the strongest; while soil organic carbon has a significant positive correlation with invertase, urease and amylase activity, with a slight influence on catalase activity. The results of this study showed that the content of organic carbon, total phosphorus and other soil fertility factors in the soil would be increased and the enzyme activity would be enhanced if the flooding degree was changed properly.

Corresponding author
Dong Cui, cuidongw@126.com

## INTRODUCTION

Wetlands are ecological systems with unique biological characteristics, soil and hydrology (*Jenkinson, Adams & Wild, 1991*). Their soil is immersed in water, and there are many different kinds of animals, plants and microorganisms with wetland characteristics. Therefore, a wetland is not only a natural landscape with affluent ecological diversity in nature, but also an important environment for human survival (*Lu & He, 2004*; *Garken et al., 2018*). According to research, water is an important environmental factor, which plays an important role in maintaining the stability of structure and function in wetland ecosystem and affecting the biogeochemical cycle in wetland (*Yu, 1999*). Therefore, in recent years, the effects of water on wetland soil factors and plant growth have been widely studied. Studies have found that plants can adapt to the stress of different flooding environments by changing their height, the stem diameter, and population density (*Tan & Zhao, 2006*). Compared with natural exposed soil, the submerged environment promoted the growth and activity of soil microorganisms, enhanced the activity of the soil enzyme (*Liu et al., 2017*). It can be seen that different water conditions have a profound influence on the growth and reproduction of wetland plants.

The Ili Valley belongs to the arid inland river basin wetland, and its total wetland area is about $2.4 \times 10^5$ km$^2$. There are many types and wide distribution of wetlands in the valley. Because of its natural environmental conditions and special geographical location, the Ili Valley has created a wetland landscape with abundant water resources and rich species. At present, the Ili Valley is a key protected biodiversity area in China (*Garken & Milligan, 2014*). In recent years, owing to human over-reclamation and the influence of natural factors, the degradation of most wetlands has become more and more serious, especially the fluctuation of water content. However, the change of water conditions will have a great impact on the process of soil carbon accumulation and decomposition (*Wana et al., 2009*). Besides, with the degradation of wetlands, the content of soil organic carbon decreased significantly, and the activity of soil enzymes also changed (*Chen et al., 2019*). There are few reports on this aspect in the Ili Valley. This study takes Ili Valley wetlands with different flooding levels as the research objects. It analyzes the change of soil enzyme activity of wetland under different flooding conditions, and it discusses the relationship between water and soil factors affecting the enzyme activity, which provided a theoretical basis for the study of the mechanism of soil water impact on soil and wetland protection in the Ili Valley.

## MATERIALS & METHODS

### Site description

The Ili River Valley is located in the northwest direction of Tian Shan Mountains in Xinjiang China, and surrounded by high mountains in the north, east and south, showing the natural geographical features of "three mountains with two valleys". It enjoys the reputation of "Wet Island in the western region" and "Jiang Nan beyond the Great Wall", and is the main transportation route of the ancient Silk Road.

In addition, the Ili Valley is situated at 80°09′E−84°56′E in the east longitude and 42°14′N−44°50 ′N in the North latitude, with an altitude of 530∼1,000 m and an area of 56,400 km². Due to the excellent natural environment and unique geographical position of the Ili Valley, the water resources and mineral resources are quite abundant, and there are various species in the valley. The climate is warm and humid, belonging to the temperate continental climate, with a great temperature difference between day and night. The annual average temperature is 10.4 °C, and the annual average sunshine hours are 2,700∼3,000 h. The annual average precipitation is approximately 417.6 mm, mainly concentrated in spring and summer, which is 60%∼70% of the annual precipitation. With the increase of altitude, the precipitation can be as high as 600 mm in mountainous areas, and the annual average evaporation is about 1,260∼1,900 mm, which is the wettest climate area in Xinjiang China.

The Ili River Valley mainly distributes forests, grasslands and meadows. The grassland soil types are mainly gray-calcium soils. The plant species are mostly perennial and cold-tolerant grasses; the forest soil is mainly taupe forest soil, and the tree species are mostly Xinjiang clouds. Cedar, snowy spruce, eucalyptus, etc (*Yang et al., 2010*).

## Study site and sample collection

The sampling sites were selected in Ili River floodplain wetland and Liberate Bridge National Wetland Park in Zhaosu County, as shown in Fig. 1. In September 2017, three kinds of wetland soils with different flooding degrees were collected in Wetland Park, i.e., *Ditch millet wetland* (DMW), *Calamagrostis epigeios wetland* (CEW) and *Phragmitesaustralis wetland* (PAW). Among them, DMW belongs to the rarely flooded habitat; CEW belongs to the seasonal flooded habitat, with a one-year flooding period of about 2∼3 months; while PAW belongs to the long-term flooded habitat, with a one-year flooding period of about 10 months.

Three plots (1 m ×1 m) were randomly set up in the selected sampling area. Firstly, in each wetland type, plant and its litter on the surface of the plot were removed with a shovel to obtain the three random soil profiles. Afterwards, soil samples of 0∼10 cm, 10∼20 cm, 20∼30 cm, 30∼40 cm were collected from bottom to top, respectively. And the total of 36 soil samples were collected from three wetland types, the collected samples were sealed in plastic bags and brought back to the laboratory. The samples were divided into two parts, one of which was stored in a sealed bag and stored in a refrigerator for the determination of soil microbial biomass carbon and soil enzyme activity, and the other was placed in a bag and air-dried, ground, and passed through a 0.15 mm sieve to determine the physical and chemical properties of the soil.

## Analysis of soil properties
### Soil physical and chemical properties

The content of soil organic carbon (SOC) was measured by a $K_2CrO_7$-$H_2SO_4$ oxidation procedure (*Lu, 1999*). The soil samples were boiled with perchloric acid and sulfuric acid. Afterwards, the total phosphorus (TP) content in soil was determined by colorimetry (*Lu, 1999*). The content of easily oxidized organic carbon (EOC) in soil can be obtained by putting potassium permanganate solution into soil sample and then colorimetric

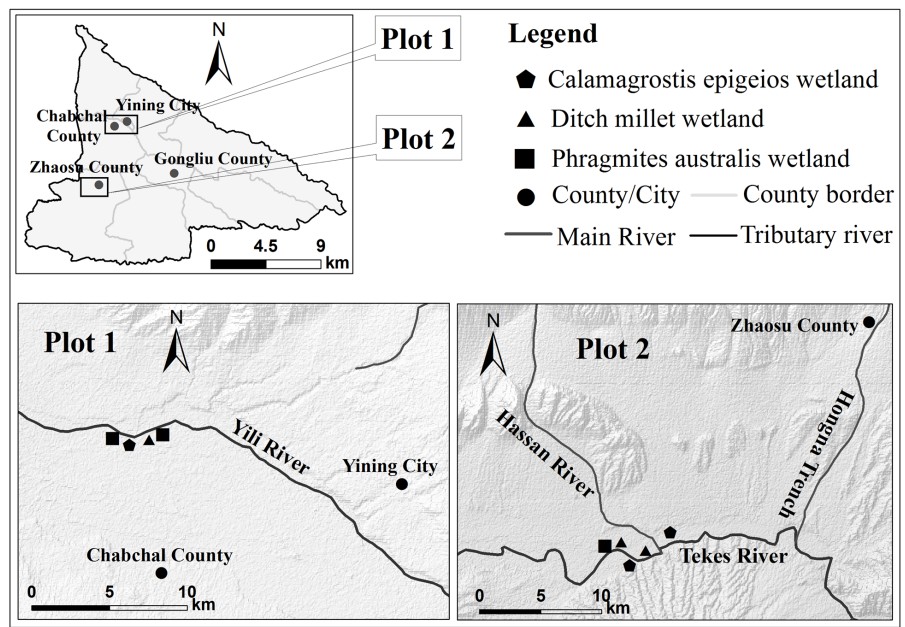

**Figure 1** **Diagram of wetland sampling point in Ili Valley.** DMW: *Ditch millet wetland*; CEW: *Calama-grostis epigeios wetland*; PAW: *Phragmites australis wetland*.

method (*Blair, Lefroy & Lisle, 1995*). The content of dissolved organic carbon (DOC) was determined by colorimetry (*Zhan & Zhou, 2002*). It is known that $NH^{4+}$ in soil leaching solution reacts with hypochlorite and phenol in strong alkaline medium to form water-soluble dye indophenol blue. The content of ammonium nitrogen ($NH^{4+}$-N) can be determined by colorimetry (*Zhao & Wang, 2011*).

### Microbial biomass carbon

The content of microbial biomass carbon (MBC) was determined by fumigation of the sample with $CHCl_3$ and extraction with 0.5 mol L-1 $K_2SO_4$ (*Vance, Brookes & Jenkinson, 1987*).

### Enzyme activity

Catalase activity was determined by measuring the $O_2$ absorbed by $KMnO_4$ in the sample added with $H_2O_2$ solution (*Rodríguez-Kábana & Truelove, 1982*). The sucrase activity was determined by measuring glucose content after incubation for 24 h at 37° C with sucrose as a substrate (*Guan, 1986*). The amylase activity was measured by colorimetry (*Zhou, 1987*), and it was determined by measuring the amount of glucose produced during hydrolysis. For the determination of urease activity (*Guan, 1986*), first one 5 g soil sample was put into a 100 ml quantificational carafe, then add 5 ml of 10% urease solution and 10 ml of citrate buffer (pH 6.7), put the quantificational carafe into incubator for 24 h at 37 °C. Finally, the released ammonium was determined colorimetrically at 578 nm using Indophenol reagent.

## Statistical analysis

The processing software (Excel 2010, SPSS 19.0 and CANOCO 4.5) were used to analyze the integrated data. One-way ANOVA method was used to analyze the differences of soil microbial biomass carbon, soil physical, chemical properties and soil enzyme activities in different flooding degrees. The two-way ANOVA method was used to analyze the degree of flooding and the depth of soil layer, and their interaction effects on soil microbial biomass carbon, soil basic physical and chemical properties and soil enzyme activity were discussed. The effects of soil physicochemical properties and microbial biomass carbon on soil enzyme activity were analyzed by RDA sorting. It should be noted that the factors significantly related to soil enzyme activities need to be selected by Monte Carlo analysis before the redundancy analysis. The T-value double sequence diagram of CANOCO can also be used to analyze the single environmental factor affecting soil enzyme activity.

# RESULTS

## Soil physical and chemical properties

The same flooding conditions, the soil physical and chemical properties of soil layers with different depths have certain differences (Table 1). Except for the PAW, the content of soil organic carbon in the DMW and CEW was obviously different among the three soil layers. The content of the SOC of the 0~10 cm soil layer of the CEW was significantly higher than that in 10~20 cm, 20~30 cm and 30~40 cm soil layers, while that in the 0~10 cm and the 10~20 cm soil layer of the DMW was significantly higher than that in the 20~30 cm and 30~40 cm soil layers.

For total phosphorus (TP), there were no significant differences between the DMW and PAW in the three soil layers, and the TP content of the CEW decreased gradually with the increase of soil depth. In terms of easily oxidized organic carbon (EOC), there were no significant differences between the CEW and PAW in the three soil layers. The EOC content in 0~10 cm soil layer was the highest, while that in 30~40 cm soil layer was the lowest in the DMW. The content of DOC in wetlands under three flooding conditions showed a decreasing trend with the increase of soil depth.

Two-way ANOVA shows that except $NH_4^+$-N, different flooding conditions and soil layers have significant effects on the soil physical and chemical properties (Table 2). Among them, the effect of flooding conditions on the physical and chemical properties of soil was greater than that of soil depth. Only the $F$ value of TP in different soil depths was higher than that of $F$ value in different flooding conditions, indicating that different soil conditions had a stronger effect on TP content. At the same time, TP and SOC are also significantly affected by the interaction between different flooding conditions and the soil depth, but the interaction has no significant effect on EOC, DOC and $NH_4^+$-N. Different flooding conditions in the same soil layer have different effects on the soil physical and chemical properties (Table 1). By comparing the average values in (Table 1), showed that the contents of TP, DOC and NH4+-N in wetlands with different flooding conditions are PAW >CEW >DMW; the EOC content in wetlands with different flooding conditions is DMW >CEW >PAW. And the SOC content in the CEW and DMW was significantly higher than that in the PAW.

**Table 1 Soil basic physical and chemical properties of different soil thickness under different flooding conditions.**

| Wetland Type | Soil Layer (cm) | SOC (g kg$^{-1}$) | TP (g kg$^{-1}$) | EOC (mg kg$^{-1}$) | DOC (mg kg$^{-1}$) | NH$_4^+$-N (mg kg$^{-1}$) |
|---|---|---|---|---|---|---|
| DMW | 0–10 | 27.17 a (7.02) | 8.15 a (0.57) | 7.39 a (0.95) | 219.30 a (3.53) | 6.99 a (0.25) |
| | 10–20 | 16.43 ab (6.42) | 7.35 a (0.50) | 4.11 b (0.85) | 199.72 ab (5.87) | 6.68 a (0.34) |
| | 20–30 | 6.57 b (2.29) | 6.61 a (0.67) | 4.63 b (0.38) | 173.28 b (6.42) | 6.59 a (0.31) |
| | 30–40 | 5.79 b (1.98) | 6.29 a (0.50) | 3.48 b (0.04) | 145.87 c (11.29) | 6.48 a (0.29) |
| CEW | 0–10 | 31.37 a (11.01) | 8.30 a (0.06) | 6.15 a (1.68) | 241.81 a (4.27) | 7.20 a (0.20) |
| | 10–20 | 15.06 ab (6.89) | 7.85 ab (0.36) | 4.09 a (0.20) | 214.40 ab (2.94) | 6.95 a (0.10) |
| | 20–30 | 11.18 ab (7.50) | 7.23 b (0.42) | 4.52 a (0.56) | 181.11 b (6.42) | 6.78 a (0.08) |
| | 30–40 | 3.32 b (0.29) | 7.20 b (0.23) | 3.25 a (1.33) | 140.00 c (20.51) | 6.76 a (0.08) |
| PAW | 0–10 | 11.66 a (5.21) | 8.72 a (0.64) | 5.07 a (3.00) | 256.50 a (36.38) | 7.22 a (0.25) |
| | 10–20 | 4.49 a (2.26) | 8.39 a (0.56) | 3.68 a (0.55) | 237.90 a (38.33) | 7.02 a (0.15) |
| | 20–30 | 1.66 a (0.49) | 7.88 a (0.88) | 3.15 a (0.79) | 206.57 a (33.46) | 6.97 a (0.24) |
| | 30–40 | 1.67 a (0.88) | 7.82 a (0.47) | 3.02 a (2.07) | 158.60 a (22.17) | 6.84 a (0.32) |

**Notes.**
The values are average (standard error). The different letters of the same column data represent significant differences among different soil layers of the same wetland ($p < 0.05$). Table 4 is the same.

**Table 2 A two-way ANOVA for the effects of different flooding conditions and soil layers on soil basic physicochemical properties.**

| Influence factor | | SOC (g kg$^{-1}$) | TP (g kg$^{-1}$) | EOC (mg kg$^{-1}$) | DOC (mg kg$^{-1}$) | NH$_4^+$-N (mg kg$^{-1}$) |
|---|---|---|---|---|---|---|
| Soil Layer | $F$ | 4.29 | 4.40 | 0.50 | 2.02 | 1.30 |
| | $P$ | 0.03 | 0.00 | 0.00 | 0.00 | 0.29 |
| Flooding Conditions | $F$ | 7.70 | 3.38 | 1.72 | 10.87 | 3.06 |
| | $P$ | 0.00 | 0.02 | 0.01 | 0.00 | 0.05 |
| Interaction | $F$ | 0.55 | 0.34 | 0.45 | 0.13 | 0.28 |
| | $P$ | 0.03 | 0.02 | 0.84 | 0.99 | 0.94 |

## Soil microbial biomass carbon

There were differences in soil microbial biomass carbon at different soil depths under the same flooding conditions (Fig. 2). The MBC content in 0~10 cm and 10~20 cm soil layers of the DMW and the CEW was significantly higher than that in 20~30 cm and 30~40 cm soil layers; the MBC content in 0~10 cm soil layers of the PAW was significantly higher

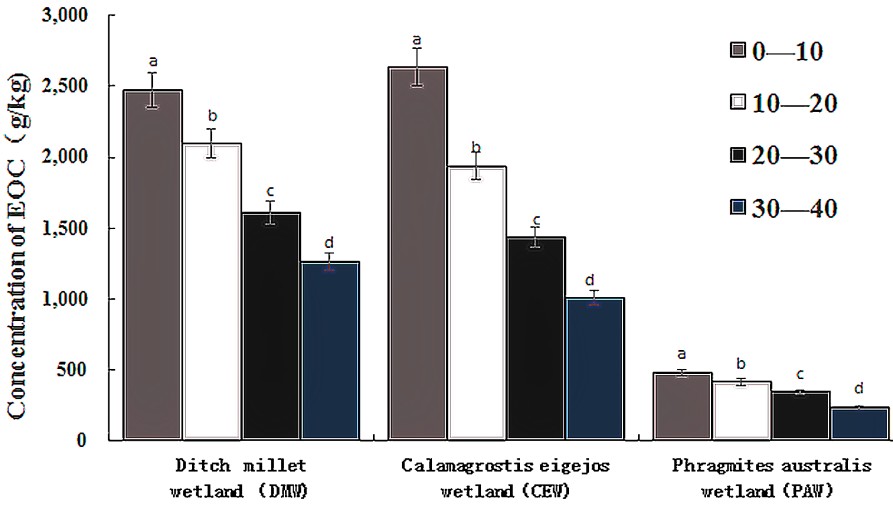

**Figure 2** Soil microbial biomass carbon content in wetland soil of different layers under different flooding conditions. .

**Table 3** A two-way ANOVA for the effects of different flooding conditions and different soil layers on soil microbial biomass carbon.

| Influence factor | | MBC |
|---|---|---|
| Soil Layer | F | 2,192.83 |
| | P | 0.00 |
| Flooding | F | 465.36 |
| Conditions | P | 0.00 |
| Interaction | F | 75.75 |
| | P | 0.00 |

than that in 30~40 cm soil layers. The MBC in the wetland with three flooding conditions showed a trend of decreasing with the increase of soil depth.

The results of two-way ANOVA showed that different flooding conditions and soil layers had significant effects on soil microbial biomass carbon, and the interaction between them also significantly affected soil microbial biomass carbon (Table 3). The content of MBC decreased gradually with the increase of flooding degree. The *F* value of MBC in different soil layers is far greater than that in different flooding conditions, which indicates that different soil depth has a deeper impact on the MBC than the flooding condition. The soil depth is indeed one of the important factors affecting the change of MBC content.

## Differences of soil enzyme activities

The activities of enzymes in different soil layers are different (Table 4). For the sucrose, the sucrase activity of wetlands with three flooding degrees decreased significantly with the increase of soil depth. Among them, the sucrase activity of 0–10 cm and 10~20 cm soil layers in the DMW and CEW was significantly higher than that of 20~30 cm and 30~40 cm soil layers; the sucrase activity of 0~10 cm soil layers in the PAW was significantly higher than that of 10~20 cm, 20~30 cm and 30~40 cm soil layers. There was no significant

**Table 4   Soil enzyme activities of different soil thickness under different flooding conditions.**

| Wetland type | Soil Layer (cm) | Sucrase (mg (g 24h)-1) | Catalase (mg/g) | Amylase (mgC6H12O6/ (g*h)) | Urease (mg/g) |
|---|---|---|---|---|---|
| DMW | 0–10 | 1.57 a (0.27) | 1.72 a (0.51) | 14.52 a (0.22) | 23.91 a (10.63) |
| | 10–20 | 1.43 a (0.31) | 1.90 a (0.30) | 6.60 b (0.23) | 13.74 a (7.38) |
| | 20–30 | 0.93 ab (0.16) | 1.91 a (0.26) | 5.75 b (0.51) | 12.01 a (7.01) |
| | 30–40 | 0.47 b (0.08) | 2.16 a (0.05) | 6.47 b (0.53) | 5.86 a (2.62) |
| CEW | 0–10 | 2.19 a (0.24) | 2.10 a (0.02) | 13.29 a (3.41) | 20.04 a (12.13) |
| | 10–20 | 1.16 b (0.36) | 2.03 a (0.06) | 8.76 ab (2.53) | 12.98 a (7.36) |
| | 20–30 | 0.44 c (0.03) | 1.96 a (0.05) | 5.84 b (0.24) | 7.96 a (2.61) |
| | 30–40 | 0.40 c (0.03) | 1.76 b (0.04) | 5.83 b (0.59) | 7.23 a (2.43) |
| PAW | 0–10 | 1.21 a (0.38) | 1.97 a (0.24) | 10.45 a (3.27) | 7.81 a (2.62) |
| | 10–20 | 0.39 b (0.03) | 1.92 a (0.25) | 6.17 a (0.81) | 6.10 a (2.28) |
| | 20–30 | 0.34 b (0.02) | 1.72 a (0.30) | 6.21 a (0.26) | 4.74 a (2.41) |
| | 30–40 | 0.32 b (0.02) | 1.17 a (0.40) | 5.92 a (0.68) | 6.86 a (3.10) |

difference in the catalase activity between the DMW and PAW, but the catalase activity decreased with the increase of soil depth in the CEW. The amylase activity of the different soil layers was not significantly different in the PAW; the amylase activity in the DMW decreased at first and then increased with the increase of soil depth; the amylase activity in the CEW decreased with the increase of soil depth, and the amylase activity in 0–10 cm soil layer was the strongest. There was no significant difference in the urease activity among the three wetlands of flooding conditions.

According to Two-way ANOVA of flooding conditions and soil depth on soil enzyme activity (Table 5). Except the sucrase, the soil depth had no significant effect on other soil enzyme activities. Different flooding conditions have significant effects on the activities of the sucrase and amylase. The $F$ values of sucrase, amylase and urease activities in different flooding conditions were higher than those in different soil depths, which indicated that the effects of different flooding conditions on soil enzyme activities were greater than those at different soil depths, and flooding conditions were one of the important factors affecting soil enzyme activities. According to the average comparison in Table 4, the activities of sucrase and urease decreased gradually with the increase of flooding degree. Among them, the activities of sucrase in the CEW significantly decreased by 4.91% compared with the

**Table 5  A two-way ANOVA for the effects of flooding conditions and soil layers on soil enzyme activities.**

| Influence Factor | | Sucrase | Catalase | Amylase | Urease |
|---|---|---|---|---|---|
| Soil Layer | F | 7.78 | 1.25 | 0.74 | 1.58 |
| | P | 0.03 | 0.30 | 0.49 | 0.23 |
| Flooding Conditions | F | 21.08 | 0.60 | 12.13 | 1.69 |
| | P | 0.00 | 0.62 | 0.00 | 0.20 |
| Interaction | F | 2.18 | 1.11 | 0.59 | 0.35 |
| | P | 0.04 | 0.38 | 0.74 | 0.90 |

DMW, and the activities of sucrase in the PAW significantly decreased by 46.04% compared with the CEW.

Although the flooding conditions had no significant effect on the catalase activity in 0~10 cm, 10~20 cm and 20~30 cm soil layers, the effects of different flooding conditions were extremely significant on the catalase activity in 30~40 cm soil layers. As far as the catalase activity in 30~40 cm soil layers was concerned, the DMW was significantly reduced by 18.45% compared with the CEW, and the CEW was significantly reduced by 33.20% compared with the PAW. The activities of sucrase and amylase in the DMW and CEW were significantly higher than those in the PAW, while the catalase activity was the highest in the CEW.

## Correlation analysis between soil enzyme activity and soil physical-chemical factors, microbial biomass carbon

Redundancy analysis (RDA) was used to analyze the relationship between soil physical and chemical factors, microbial biomass carbon and soil enzyme activities in wetlands under different flooding conditions (Fig. 3). The results showed that the first two sorting axis together explained 52.6% of the change of soil enzyme activity, of which the contribution rate of the first sorting axis (RDA 1) was 47.6% and that of the second sorting axis (RDA 2) was 5%. This indicated that most of the information between soil physical and chemical factors, microbial biomass carbon and soil enzyme activities could be reflected by these two axes, and was mainly determined by the first sorting axis. According to the redundancy analysis (Fig. 3), the arrow lines of SOC, DOC and TP are the longest, which together with the importance sorting results of Table 6 shows that SOC, DOC, TP and MBC can explain the changes of soil enzyme activities very well. The angles are small and the directions are the same between SOC and the sucrose, TP and catalase, which indicates that there are significant positive effects between SOC and the sucrase activity, TP and the catalase activity. SOC may be the dominant factor affecting the sucrase activity in Ili Valley, and TP is an important factor affecting the catalase activity.

A single environmental factor analysis was carried out for the environmental factors affecting soil enzyme activity by using the T-value double-sequence diagram of CANOCO 4.5 (Fig. 4). As shown in Fig. 4A, the arrows of sucrase, urease and amylase all fall on the solid line circle of SOC, indicating that SOC has a significant positive correlation with the sucrase, urease and amylase activities, that is to say, the activities of sucrase, urease and

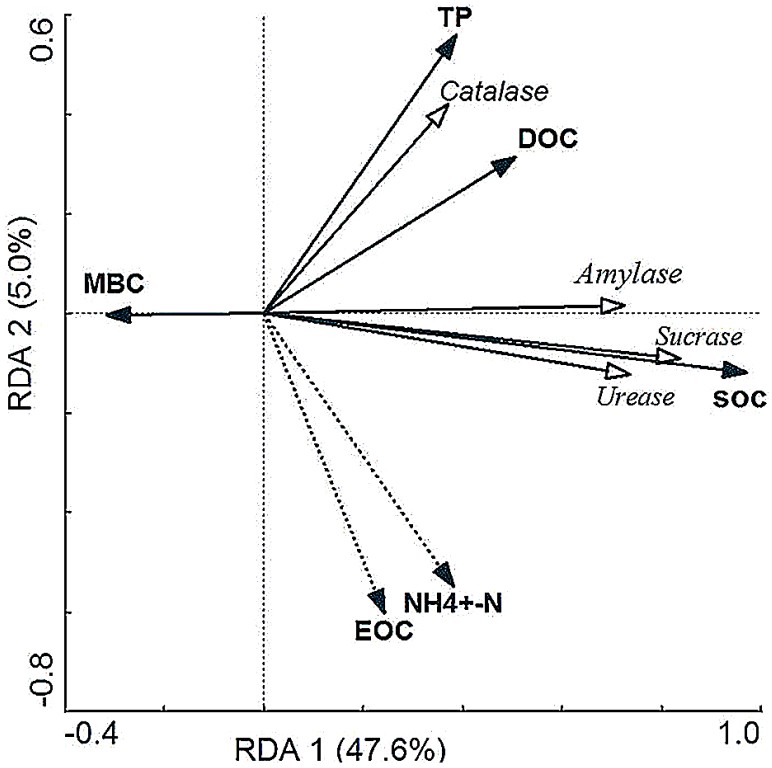

**Figure 3** **Redundancy analysis of the effect of soil physical and chemical properties on soil enzyme activity.** The quadrant of arrow in the figure represents the positive and negative correlation between different factors and the sorting axis, the hollow arrow represents several soil enzymes, the solid arrow represents environmental variables, and the cosine value of corresponding angle represents the correlation between environmental variables and soil enzymes. With the smaller the cosine value, the greater the correlation. Solid line represents the factors significantly related to the soil enzyme activity ($p < 0.05$).

**Table 6** **Significance rank and significance test of soil physicochemical factors and microbial biomass carbon in explanation.**

| Environmental factor | Sorting of importance | Degree of interpretation (%) | Importance (*F* value) | Significance (*P* value) |
|---|---|---|---|---|
| SOC | 1 | 45 | 27.824 | 0.002 |
| DOC | 2 | 13.3 | 5.209 | 0.012 |
| TP | 3 | 8.9 | 3.337 | 0.032 |
| MBC | 4 | 5.1 | 2.728 | 0.050 |
| $NH_4^+$-N | 5 | 8.5 | 3.139 | 0.114 |
| EOC | 6 | 4.7 | 1.284 | 0.296 |

amylase increase with the increase of SOC content. The arrow of catalase passes through the solid line circle of SOC, which shows that there is a positive correlation between SOC and CAT. Figure 4B shows that four soil enzymes pass through the solid line circle of DOC, which indicates that there are a positive correlation between DOC and the activities of

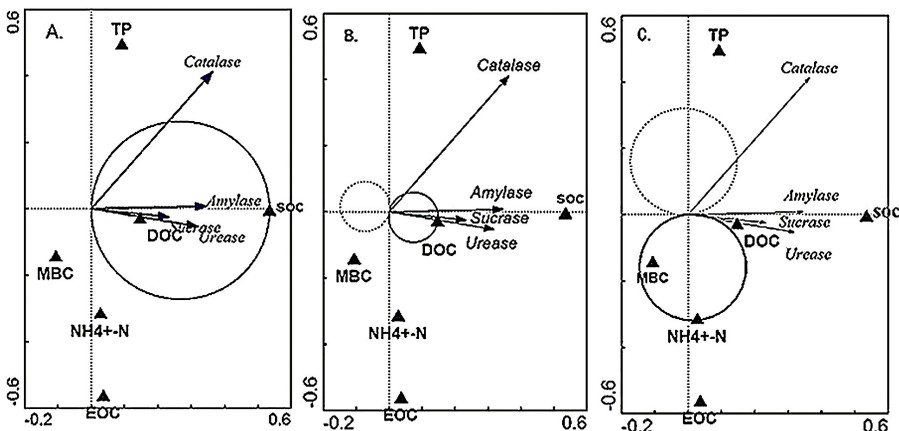

**Figure 4 The T-value for a single factor influencing varied of soil enzyme activities in wetland.** The quadrant of the arrow in the figure represents the positive and negative correlation between different factors and the sorting axis. The arrow represents several soil enzymes, and the solid triangle represents environmental variables. (A) The single environmental factor analysis of SOC; (B) the single environmental factor analysis of DOC, (C) the single environmental factor analysis of $NH^{4+}$-N.

the four soil enzymes. Figure 4C shows that most of the four soil enzymes fall outside the solid line circle and dotted line circle of $NH^{4+}$-N, indicating that there was no significant relationship between $NH^{4+}$-N and the activities of the four soil enzymes.

Different environmental factors have different effects on soil enzyme activity (Table 6). The effects of different environmental factors on soil enzyme activities were SOC >DOC >TP >MBC >$NH^{4+}$-N >EOC. Among them, the effects of SOC, DOC, TP and MBC were significant on soil enzyme activity, especially the effect of SOC on soil enzyme activity was extremely significant. And SOC had the greatest effect on soil enzyme activity, accounting for 45% of the total explanations ($F = 27.82$, $p < 0.01$). The effects of $NH^{4+}$-N and EOC were not significant on soil enzyme activity ($p > 0.05$).

## DISCUSSION

### Effects of different flooding conditions on soil enzyme activities

Soil enzyme is a kind of proteins with special catalytic ability, which mainly comes from the decomposition of soil microorganisms, animal and plant secretions and residues (*Guan, 1986*). Soil moisture has a significant correlation with soil microbial activity and type, and different water conditions will directly affect the existence and activity of soil enzyme activity (*Wan & Wu, 2005*). It indicates that the degree of flooding did significantly affect the activity of soil enzymes. The results showed that the activities of sucrase and amylase were closely related to the degree of flooding, and decreased gradually with the increase of flooding degree (*Wan et al., 2008*; *Zhou, 2018*), which was consistent with the effect of flooding degree on sucrase and amylase activities, and the activities of sucrase and amylase in the rarely flooded and seasonal flooded wetlands were significantly higher than those in the long-term flooded wetlands. The phenomenon may be due to the increase of soil moisture, which leads to the decrease of soil permeability, restricts the growth of soil

microorganisms, greatly slows down the decomposition of soil humus, and thus reduces the activity of soil enzymes. Studies have shown that soil moisture can affect soil microbial biomass by changing soil oxygen content (*Rousk, Brookes & Erland, 2009*). Therefore, it is also possible that in an environment with sufficient substrate and moist soil, increasing soil water will affect the availability of oxygen, thus affecting the growth of soil microorganisms and plant roots, resulting in the decrease of enzyme activity (*Guenet et al., 2012*).

In this study, although there is no significant correlation between flooding conditions and catalase, urease activity, as a whole, urease activity in very few flooded wetlands is much higher than that in perennial flooded wetlands, which may be because the deeper the soil layer in the wetland, the less the water content, the more conducive to the accumulation of soil organic matter and the improvement of enzyme activity. This is similar to the results of other researchers (*Xu et al., 2017*). In this study, the catalase activity did not change significantly with the increase of the water content, which was consistent with the result that the catalase activity did not change significantly with the water gradient in non-rhizosphere soil (*Tian et al., 2012*).

## Effects of microbial biomass carbon and soil physical-chemical properties on soil enzyme activities

Soil microbial biomass is an essential indicator for soil quality. On the one hand, it is highly sensitive and reflects small changes in soil before total carbon changes (*Powlson, Brookes & Christensen, 1987*). On the other hand, it also reflects the process of nutrient transfer and the energy cycle (*Doran, 1994*). Previous studies have shown that the carbon contents of soil microbial biomass and soil enzyme activities in the treatment of underwater are higher than those in the treatment of natural bareness (*Liu et al., 2017*). (Table 6) shows that soil microbial biomass carbon is significantly correlated with soil enzyme activity, which is closely related to soil microbial biomass carbon and soil enzyme activity obtained by the predecessors. With the change of water gradient, soil enzyme activity is positively correlated with microbial biomass carbon (*Wan et al., 2008*). This is significantly related to the microbial biomass carbon and soil enzyme activity in this study, and the results are consistent with the law of decrease with the increase of water content (Table 6). This phenomenon is due to the fact that the soil aeration of rarely flooded wetlands is good, the vegetation grows luxuriantly, accelerates the decomposition of soil humus, provides a large number of carbon sources for the metabolism process of soil microorganisms, and leads to the increase of microbial biomass carbon (*Jia et al., 2005*).

There was a certain relationship between soil organic carbon and soil enzyme activity (*Wan et al., 2009*). Soil organic matter is the main source of enzyme substrate, and the content of organic matter greatly affects the activity of soil enzyme. It can be seen from Table 4 and showed that the activities of sucrase and amylase basically decrease with the increase of soil depth. This may be because plant roots are mostly concentrated on the surface of the soil. Vigorous root activity promotes the turnover of litter on the ground. The high content of organic matter in the soil leads to the increase of the respiration intensity of microorganisms, which is convenient for the reproduction of microorganisms, so that the higher soil enzyme activity is accumulated on the surface of the soil (*Wan & Song,*

*2008*). It can be seen from Fig. 3 that there is a significant positive correlation between soil organic carbon and sucrase, urease and amylase activities; there is a positive correlation between soil oxidizable organic carbon and sucrase, urease, catalase and amylase activities. This shows that most of the soil enzymes are sensitive to the change of soil active organic carbon composition, which is consistent with the previous research results (*Xiao et al., 2015*). The soil organic matter in the degraded wetland of Napa sea in Northwest China changed from original swamp to swamp grassland, the cultivated land decreased gradually, and the activity of soil enzyme showed the same trend (*Lu et al., 2004*; *Lu & He, 2004*). These results indicate that the soil enzyme activity is closely related to soil nutrients such as soil organic matter, and the change of soil enzyme activity can better reflect the degree of soil degradation.

In this study, total phosphorus has a significant positive effect on the activities of four soil enzymes, and has the strongest correlation with the activities of catalase, which can better explain the changes of catalase activity, indicating that total phosphorus is the main factor affecting the activities of urease, catalase and sucrase through direct or indirect effects (*Liu et al., 2003*). Previous studies have shown that the relationship between ammonium nitrogen and soil enzyme activity is not significant, and with the increase of soil depth, it first decreases and then increases (*Xu et al., 2017*). The content of oxidized organic carbon in different wetland types decreased with the increase of soil depth (*Xiao et al., 2015*). This is similar to the conclusion that the relationship between ammonium nitrogen, easily oxidized organic carbon and soil enzyme activity is not significant, and decreases with the increase of soil depth. This may be because the increase of water content can adjust the physical structure of soil, improve the effectiveness of soil nutrients, and facilitate the transfer of easily oxidized organic carbon and ammonium nitrogen in the soil (*Wang et al., 2014*). It can be seen that the decrease of wetland water will lead to the loss of soil nutrients, which will cause a large area of wetland degradation.

## CONCLUSIONS

The results show that soil organic carbon (SOC), dissolved organic carbon (DOC) and total phosphorus (TP) had significant effects on soil enzyme activity in wetland, while ammonium nitrogen ($NH_4^+$-N), easily oxidized organic carbon (EOC) and microbial biomass carbon (MBC) had no significant effects on soil enzyme activity. Among them, the correlation between soil organic carbon and the sucrase activity, total phosphorus and the catalase activity were the strongest, indicating that soil organic carbon is the main factor affecting sucrase activity, and total phosphorus is an important factor affecting catalase activity. Soil organic carbon had a significant positive correlation with sucrase, urease and amylase activity, but had a slight influence on catalase activity. Dissolved organic carbon had a positive correlation with four soil enzyme activities. It can be seen that the activity of soil enzyme in wetland is related closely to soil organic carbon and dissolved organic carbon.

Compared with the soil enzyme activities, it was found that the activities of sucrase and urease in wetland were in the order of rarely flooded wetlands >seasonal flooded wetlands

>long-term flooded wetlands with the increase of flooding degree. The activity of amylase in rarely flooded wetland and seasonal flooded wetlands was significantly higher than that in long-term flooded wetlands. The activity of catalase in seasonal flooded wetlands was the highest. All of these indicate that the humid environment will inhibit the survival of plants and microorganisms in the soil, hinder the decomposition of organic matter, and lead to the decrease of enzyme activity. In conclusion, the soil enzyme activity is closely related to soil nutrients such as soil organic matter and water, and the change of soil enzyme activity can better reflect the degree of soil degradation.

## ACKNOWLEDGEMENTS

The authors would like to thank Dr. Cui Dong, Dr. NijatKasim and Dr. Yang Haijun for his suggestions on sample collection in the early experimental stage, as well as for the revision and supplement of the manuscript. Thanks are also given for the collection of materials by San Sancai, Gu Gie and Niu Mengmeng, and the map of the wetland sampling sites made by Dr Yan Junjie. Dr. Nijatkasim is thanked for his help on language issues in the manuscript.

### Funding

This work was supported by the Tianshan Youth Program, a special talent program in Xinjiang Uygur Autonomous region (No.2018Q076). The funders had no role in study design, data collection and analysis, decision to publish, or preparation of the manuscript.

### Grant Disclosures

The following grant information was disclosed by the authors:
Tianshan Youth Program, a special talent program in Xinjiang Uygur Autonomous region: No.2018Q076.

### Competing Interests

The authors declare there are no competing interests.

### Author Contributions

- Yulu Zhang conceived and designed the experiments, performed the experiments, analyzed the data, prepared figures and/or tables, authored or reviewed drafts of the paper, and approved the final draft.
- Dong Cui conceived and designed the experiments, performed the experiments, analyzed the data, authored or reviewed drafts of the paper, and approved the final draft.
- Haijun Yang conceived and designed the experiments, performed the experiments, authored or reviewed drafts of the paper, and approved the final draft.
- Nijat Kasim conceived and designed the experiments, performed the experiments, authored or reviewed drafts of the paper, revised the language problems, and approved the final draft.

## Field Study Permissions

The following information was supplied relating to field study approvals (i.e., approving body and any reference numbers):

No approval from any department is required to enter the sampling place where the experiment is located, because the selected sampling place is a free and open public place, allowing the data required for the experiment to be collected in the sampling place. At the same time, the sample collection does not involve any drugs, and all the experimental operations will not damage the local natural conditions and soil environment, or even worse.

## Data Availability

The raw measurements are available in the Supplementary File.

## Supplemental Information

Supplemental information for this article can be found online at http://dx.doi.org/10.7717/peerj.8531#supplemental-information.

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
