# Peer review of "Differences of soil enzyme activities and its influencing factors under different flooding conditions in Ili Valley, Xinjiang"

_PeerJ, doi:10.7717/peerj.8531_

## Round 0.1 · original submission · Major Revisions

Dear Dr. Zhang,

Due to the number of different issues raised by the three reviewers, I think the manuscript needs a substantial revision before I accept it for publication in PeerJ. Therefore if you will be willing to consider and send me back a letter with every issue discussed I will then consider for sending it back to one of the reviewers to get its feedback.

I hope you can work on it because it has a good potential to be published here after the revision.

Best regards,

Gabriela Nardoto

·

Basic reporting

THE paper is a primary data collection,in which authors discuss about water roles under different types of conditions. More revisions are needed to focus on the given target by good logic and recent references.

Experimental design

ok

Validity of the findings

ok

Additional comments

The paper needs major revision in terms of writing,more recent citations and concrete conclusions and recommendations.

Reviewer 2 ·

Basic reporting

This paper is interesting and brings important subject. However, some points should be addressed. The English must be corrected. Some parts of paper are not clear and with no sense. Please check the paper carefully and send it to an English native service.

Experimental design

No comment

Validity of the findings

Wetland is a topic of interest globally, this paper deals with an interesting research area.
The objective is fulfilled, since a very meticulous analysis of the soil microbial biomass and enzyme activity. The work was made with precise and reliable techniques.
It is a work of great importance that generates very valuable information especially for future analyzes that give an account of the state of health of the soils wetlands regions.

Additional comments

This paper is interesting and brings important subject. However, some points should be addressed.

Abstract
I suggest rewriting the results presentation, it's too long and hard to understand.
I suggest a conclusion in the Abstract.
Line 18 - Water is an important component of wetland.
This sentence isn´t necessary.
Line 48 and 49 - Keywords: Yili valley; wetland; flooding conditions; differences in enzyme activity; microbial biomass carbon.
Remove differences in

Introduction

Line 77 - The name of the valley is incorrected.

Line 81- soil is with lowercase letter

Line 106 - Sampling point setting and field sample collection
I suggest change this section to “Study site and sample collection”.

Line 111-113 - The flooding period of one year is about 2-3 months, while PAW belongs to long-term flooded habitat, and the flooding.
Which months? Year? Precipitation data.

Line 116 – How soils samples were collected?

Line 126 – Research method
I suggest change this section to “Laboratory analysis or Analysis of soil properties”

Line 127 - Soil basic physical and chemical properties
Remove the word basic.

Line 128 – I suggest: "Soil organic carbon (SOC) was analyzed or determined by REFERENCE method."

Line 138 – Remove the word soil.

Line 139- I suggest: "Microbial biomass C was analyzed or determined REFERENCE method."

Line 144 - Remove the word soil

Line 145 - I suggest: " Urease, catalase, sucrase activities were analyzed according to the methodology used previously by REFERENCES”.

Line 161- Remove the word basic.

Results

I felt that the authors could have explored the results more and discussed their findings more thoroughly. Particularly, I believe that the authors should highlight how their results compare to other wetland studies.
Examples: Shao et al. 2015 (Seasonal Dynamics of Soil Labile Organic Carbon and Enzyme Activities in Relation to Vegetation Types in Hangzhou Bay Tidal Flat Wetland - doi.org/10.1371/journal.pone.0142677); Ma et al. 2018 (Bacterial and Fungal Community Composition and Functional Activity Associated with Lake Wetland Water Level Gradients - doi: 10.1038/s41598-018-19153-z); Gu et al. 2019 (Soil Enzyme Activity in Soils Subjected to Flooding and the Effect on Nitrogen and Phosphorus Uptake by Oilseed Rape - https://doi.org/10.3389/fpls.2019.00368)
Line 202 – Remove “(MBC) under different flooding conditions”.

Line 221 - Remove “in wetlands under different flooding conditions”.

Line 332 – I suggest a review about this reference because this paper (Powlson et al. 1987) it is not specific about this subject. The reference can be used in the discussion in another way.

Conclusions
Conclusions should be improved.

References
References format needs to be revised.

Line 445 – This reference needs to be corrected

Table 5 – correct the legend

Figure 3 - correct the legend

Reviewer 3 ·

Basic reporting

The manuscript needs to be revised thoroughly, in which a language advisor should be involved.

Experimental design

No comments

Validity of the findings

No comments

Additional comments

1) The first part of material and methods, which gave depiction on Yily valley should be integrate into Introduction;
2) Sampling site has been showed in a map, however, this is no coordinate. These data should be provide.
3) As stated in the introduction, there is a risk of soil degradation threatening Yily Valley eco. However, in the discussion part, the results have not been discussed together with any aspect overlapping with the risk;
4) There are some advises in attachment.

Annotated reviews are not available for download in order to protect the identity of reviewers who chose to remain anonymous.

---

## Round 0.2 · accepted · Accept

Dear Dr. Zhang,

Your article was returned to one of the previous reviewers and based on their review and the quality of the revised manuscript, I'm confident this will make a good contribution to this field of knowledge.

All the best,

Gabriela Nardoto

Reviewer 2 ·

Basic reporting

The authors made corrections as suggested, which made the article more complete and easy to read.

Experimental design

All correction suggestions have been accepted, the paper is now complete.

Validity of the findings

The authors made corrections as suggested, which made the paper complete.

Additional comments

I would like to say that for my part no compliment is needed in its correction.
Congratulations to the authors for their dedication.